

# Conditioned culture medium of bone marrow mesenchymal stem cells promotes phenotypic transformation of microglia by regulating mitochondrial autophagy

Hangyu Ji[1,2], Weiming Chu[3], Yong Yang[3], Xin Peng[2] and Xiaoli Song[4]

[1] The Department of Orthopedics of ZhongDa Hospital, Southeast University, Nan Jing, Jiangsu Province, China
[2] School of Medicine, Southeast University, Nanjing, Jiangsu Province, China
[3] The Department of Orthopedics, Xishan People's Hospital, Wuxi, Jiangsu Province, China
[4] College of Chemistry and Chemical Engineering, Yangzhou University, Yangzhou, Jiangsu Province, China

## ABSTRACT

**Objective**. To study the mechanism by which conditioned medium of bone marrow mesenchymal stem cells (BMSCs-CM) facilitates the transition of pro-inflammatory polarized microglia to an anti-inflammatory phenotype.

**Methods**. BV2 cells, a mouse microglia cell line, were transformed into a pro-inflammatory phenotype using lipopolysaccharide. The expression of phenotypic genes in BV2 cells was detected using real-time quantitative PCR (RT-qPCR). Enzyme-linked immunosorbent assay was used to measure inflammatory cytokine levels in BV2 cells co-cultured with BMSCs-CM. The expressions of mitophagy-associated proteins were determined using western blot. The mitochondrial membrane potential and ATP levels in BV2 cells were measured using JC-1 staining and an ATP assay kit, respectively. Additionally, we examined the proliferation, apoptosis, and migration of C8-D1A cells, a mouse astrocyte cell line, co-cultured with BV2 cells.

**Results**. After co- culture with BMSCs -CM, the mRNA expression of tumor necrosis factor-α (TNF-α) and inducible nitric oxide synthase significantly decreased in pro-inflammatory BV2 cells, whereas the expression of CD206 and arginase-1 significantly increased. Moreover, TNF-α and interleukin-6 levels significantly decreased, whereas transforming growth factor-β and interleukin-10 levels significantly increased. Furthermore, co-culture with BMSCs-CM increased mitophagy-associated protein expression, ATP levels, mitochondrial and lysosomal co-localization in these cells and decreased reactive oxygen species levels. Importantly, BMSCs-CM reversed the decrease in the proliferation and migration of C8-D1A cells co-cultured with pro-inflammatory BV2 cells and inhibited the apoptosis of C8-D1A cells.

**Conclusion**. BMSCs-CM may promote the transition of polarized microglia from a pro-inflammatory to an anti-inflammatory phenotype by regulating mitophagy and influences the functional state of astrocytes.

Corresponding authors
Hangyu Ji, ji_hang_yu@163.com
Xiaoli Song, xlsong@yzu.edu.cn

## INTRODUCTION

Spinal cord injury (SCI) is a severe neurological condition that causes permanent and degenerative damage to the central nervous system (CNS), which in turn leads to loss of motor and sensory function (*Clifford et al., 2023*). Although a variety of treatments for SCI have been developed, none have produced meaningful functional recovery after injury (*Hellenbrand et al., 2021*). Stem cell transplantation is considered a promising approach for acute spinal cord injury, and bone marrow mesenchymal stem cells (BMSCs) are the most commonly used stem cells due to their notable advantages. A large number of studies have confirmed that transplanted BMSCs are indeed effective in SCI treatment. However, research has shown that only about 1% of BMSCs survive 1 week post-implantation *in vivo*. Studies have indicated that the main reason for their effectiveness lies in their secreted cytokines and paracrine reactions. The fluid excreted with paracrine activity is called conditional medium (CM) (*Cantinieaux et al., 2013*). CM has the same effect as BMSC transplantation, with the advantage that it does not have the limitation of stem cells (*Tsai et al., 2018*).

In SCI research, the shift in microglia phenotype is considered an important strategy (*Li et al., 2023*; *Su et al., 2020*). Microglia are macrophages resident in the CNS that help maintain CNS homeostasis through continuous interactions with neuronal and non-neuronal cells (*Brennan et al., 2022*). Activated microglia are the main cell types that trigger neuroinflammatory responses in SCI. Based on their function, activated microglia can be classified into pro-inflammatory and anti-inflammatory phenotypes. While anti-inflammatory microglia repair damaged CNS tissue, pro-inflammatory microglia cause damage to neurons, astrocytes, and vascular endothelial cells (*Brennan & Popovich, 2018*). Therefore, promoting the transition from a pro-inflammatory to an anti-inflammatory microglial phenotype is beneficial to protect the recovery of nerve function after SCI.

The formation of glial scar after SCI is a key factor that hinders the recovery of nerve function and is mainly regulated by reactive astrocytes. Astrocytes are the most common glial cells in the brain that play important roles in brain homeostasis (*Oksanen et al., 2019*), such as regulating blood flow, maintaining the blood–brain barrier, regulating synaptic activity, controlling neurotrophic factor secretion, clearing dead cells, and regulating scar formation (*Colombo & Farina, 2016*; *Kwon & Koh, 2020*). In the early stage of SCI, reactive glial cells increase and recruit other cells, such as fibroblasts, to promote the formation of glial scars in the injured area and protect the invasion of harmful substances in the injured tissue, whereas in the later stage of SCI, stable glial scars hinder the growth of neuronal axons, resulting in the loss of motor and sensory functions (*Clifford et al., 2023*).

Crosstalk between microglia and astrocytes is involved in the occurrence of many neuroinflammatory and neurodegenerative diseases (*Hasel et al., 2023*). By releasing different signaling molecules, microglia and astrocytes establish autocrine feedback and engage in a two-way dialogue to tightly regulate each other during CNS injury. Microglia not only regulate the innate immune function of astrocytes but also determine the function of reactive astrocytes, ranging from neuroprotection to neurotoxicity (*Jha et al., 2019*). Inflammatory mediators secreted by pro-inflammatory microglia, such as interleukin

(IL)-1α, IL-1β, and tumor necrosis factor-α (TNF-α), activate reactive astrocytes and induce secondary inflammatory responses (*Liddelow et al., 2017*).

In this study, lipopolysaccharide (LPS) was first used to induce the transition of BV2 cells, a mouse microglia cell line, to a pro-inflammatory phenotype. Subsequently, the effects of BMSCs conditioned medium (BMSCs-CM) on the phenotypic transformation of pro-inflammatory BV2 cells and possible regulatory mechanisms were investigated by co-culture, western blotting, cell loss assays, real-time quantitative PCR (RT-qPCR), enzyme-linked immunosorbent assay (ELISA), and other methods. Finally, we analyzed whether the transformed microglia could regulate the functional activities of astrocytes. The study forms the basis for future SCI *in vivo* experiments and provides a theoretical foundation for potential clinical treatments.

# MATERIALS AND METHODS

## Cell culture and preparation of BMSCs-CM

BV2 microglia and C8-D1A astrocyte cell lines were purchased from KeyGEN BioTECH Co., Ltd (Nanjing, China). The cells were cultured in high-glucose Dulbecco's modified Eagle's medium (DMEM; KeyGEN) containing 10% fetal bovine serum (Gibco, Waltham, MA, USA) and 1% penicillin-streptomycin (P/S; KeyGEN), in an atmosphere of 5% $CO_2$ and 95% humidity at 37 °C.

Immortalized mouse BMSCs (iCell Bioscience Inc., Shanghai, China) were cultured in a medium containing complete mesenchymal stem cell culture solution (iCell Bioscience Inc.) at 5% $CO_2$, 95% humidity, and 37 °C. When the fusion degree of BMSCs reached 70–80%, the culture was continued with serum-free medium for 48 h. Thereafter, the medium was collected and centrifuged at 3,000 g to obtain BMSCs-CM. The experiments in this study were conducted before and after CM intervention. The number of each experimental group was 3.

## RNA extraction and RT-qPCR

RT-qPCR was used to detect the expression of related genes in BV2 cells. The procedure was conducted following the instructions provided for the RT-qPCR and first cDNA synthesis kits (RR036B; TaKaRa, Shiga, Japan), as outlined below: The fully lysed BV2 cell fluid was transferred to a 1.5 mL centrifuge tube, and chloroform was added. After centrifugation, the colorless upper water phase was absorbed and transferred to another centrifuge tube. After centrifugation, equal volumes of isopropyl alcohol and 70% ethanol were added, and RNase-free water was added to dissolve the RNA precipitation. After determining the concentration and purity of RNA, a sterilized 0.2 mL PCR tube without nuclease was taken, RNA (2 μg) and 5 × PrimeScript RT Master Mix were added successively, and the product was obtained by centrifugation. The cDNA sample was diluted, and 2 × Real-time PCR Master Mix (SYBR Green, RR086B; TaKaRa, Shiga, Japan), template, and primer MIX were added into 0.1 mL PCR tube successively.

## ELISA

The effects of BMSCs-CM on pro-inflammatory (TNF-α and IL-6) and anti-inflammatory (TGF-β and IL-10) factors of BV2 cells were detected using ELISA. The tests were performed

according to the instructions of each kit (Shanghai Coaibo Biotechnology Co., Ltd., Shanghai, China). The absorbance of each well was measured at 450 nm using Spark multifunctional enzyme marker (TECAN, Männedorf, Switzerland).

## Cell proliferation assay

The proliferation of astrocytes was detected using the EdU method. The tests were conducted following the instructions provided on the EdU kit (KGA337; Jiangsu Kaiji Biotechnology Co., Ltd., Jiangsu, China). The procedure was as follows: C8-D1A cell suspension was added to the lower chamber of the co-culture plate, BV2 was inoculated in the upper chamber, and LPS and BMSCs-CM were added to the upper chamber according to the respective groups. The culture plate was incubated in a 5% $CO_2$ incubator at 37 °C for 48 h. After discarding the medium in the lower chamber, the wells were washed twice with PBS. Then, Edu (200 µM in medium preparation) was added, followed by adding 200 µL PBS containing 4% paraformaldehyde to each well. Subsequently, 200 µL of 2 mg/mL glycine and 500 µL PBS were added in turn, followed by washing with a decolorizing shaker for 5 min and discarding PBS. The prepared 1 ×-Hoechst 33342 reaction solution was added with 100 µL of 1 × Apollo staining reaction solution and penetrant, and 100 µL of 1 × Hoechst 33342 reaction solution was added to each well. After incubating for 30 min in a decolorizing shaker at room temperature and in dark, the high content cell imaging system was used for detection (magnification: 200×).

## Flow cytometry analysis

Flow cytometry was conducted to detect the apoptosis of C8-D1A cells. The tests were performed according to the instructions of the cell apoptosis detection kit (KGA105; Jiangsu Kaiji Biotechnology Co., Ltd., Jiangsu, China). The method was as follows: Cells in the logarithmic growth stage were inoculated into 6-well plates, and corresponding culture medium was added according to the group settings. At the same time, a negative control group was set up. The cells were digested and collected with 0.25% pancreatic enzyme (excluding EDTA), washed with PBS twice (centrifugation at 1,000 rpm; $5 \times 10^5$ cells were collected), and suspended in 500 µL binding buffer. Then, 5 µL Annexin V-FITC and 5 µL propidium iodide were added, and the reaction mixture was incubated at room temperature in the dark for 5–15 min.

Changes in the mitochondrial membrane potential of BMSCs-CM treated inflammatory BV2 cells were detected using JC-1 staining. The tests were performed according to the instructions on the mitochondrial membrane potential detection kit (KGA602; Jiangsu Kaiji Biotechnology Co., Ltd.). The method was as follows: cells in the logarithmic growth stage were digested and inoculated into the 6-well plate. After the cells were glued to the wall, the corresponding drug-containing medium (concentration: 5 µM) was added according to the group setting. After 48 h, the cells were digested, collected, and washed with PBS. Subsequently, 500 µL of JC-1 working liquid was added, and the cells were uniformly suspended and centrifuged at room temperature (1,000 rpm, 5 min). The cells were collected, and 500 µL 1× incubation buffer was added to the cells. Then, the cells were re-suspended and tested.

## Wound healing assay

Migratory changes of C8-D1A cells were detected using the scratch method as follows: C8-D1A cells in the logarithmic growth stage were digested and inoculated into the lower chamber of the co-culture plate, and BV2 cells were inoculated into the upper chamber. The following day, after the cells had adhered to the wall, cell aggregation in the lower chamber reached approximately 80%. LPS and BMSCs-CM were added to the upper chamber according to the group settings, with a negative control group also included. After co-culture for 48 h, photos (magnification: 100×) were taken to measure the cell migration distance.

## Western blotting

Western blotting was conducted to detect mitophagy-related proteins of BV2 cells, following the instructions provided with the whole protein extraction, BCA protein content detection, and western blotting detection kits (KGP250, KGA902, and KGP1201, respectively, Jiangsu Kaiji Biotechnology Co., Ltd.). The procedure was as follows: pro-inflammatory BV2 cells were collected, lysate was added, protein samples were extracted, and protein concentration was calculated according to the standard curve. A 10% SDS-PAGE gel was prepared for electrophoresis, after which the gel was transferred to a PVDF membrane and closed with 1% BSA for 1 h. Primary antibodies Parkin, LC3B, and P62 were then added and incubated at 4 °C overnight. Subsequently, G:BOX Chemi XR5 was added and incubated at room temperature for 2 h. The obtained strips were analyzed for grayscale using Gel-Pro32 software.

## Reactive oxygen species (ROS) assay

Inflammatory BV2 cells were seeded into confocal small dishes at a density of approximately $5 \times 10^4$ cells per dish. Following cell attachment, LPS and BMSCs-CM were added as required, and staining was performed after 48 h. The culture medium was then removed and washed with PBS three times. Subsequently, 300 µL of a diluted ROS fluorescent probe (1:1000; KGAF018; Jiangsu Kaiji Biotechnology Co., Ltd.) was added to the cells, which were then incubated for 30 min. Following this, 300 µL of Hoechst 33342 staining solution (KGA212-10; Jiangsu Kaiji Biotechnology Co., Ltd.) was added and incubated at room temperature for 10 min in the dark. The cells were then washed with PBS three times and observed under a confocal laser microscope.

## ATP chemiluminescence assay

Mitochondrial metabolism of BV2 cells was assessed using the ATP content chemiluminescence method. The procedure was performed according to the instructions provided with the ATP chemiluminescence assay kit (E-BC-F002; Wuhan Ellerite Biotechnology Co., Ltd.). Briefly, BV2 cells were collected, and 0.3 mL of reagent 1 (in a two mL EP tube) was added for every $2 \times 10^6$ cells. The tubes were tightly covered, thoroughly mixed, and then placed in a boiling water bath for 10 min. After cooling the water to 4 °C, the samples were centrifuged at 10,000× g for 10 min. The supernatant was collected, diluted, and subjected to detection using the chemiluminescence detector.

## Immunofluorescence

BV2 cells were digested and seeded into confocal small dishes at a density of approximately $5 \times 10^4$ cells per dish. Following cell adhesion, LPS and BMSCs-CM were added, and the cells were incubated for 48 h. The medium was then removed, and the cells were washed with PBS three times. Subsequently, 300 µL of a diluted Mito-Tracker Green probe and Lyso-Tracker Red probe (1:1000; KGMP007/KGMP006; Jiangsu Kaiji Biotechnology Co., Ltd.) were added to the cells and incubated for 30 min. After washing with PBS three times, 300 µL of Hoechst 33342 dyeing solution (KGA212-10; Jiangsu Kaiji Biotechnology Co., Ltd.) was added and incubated at room temperature for 10 min in the dark. The cells were then washed with PBS three times and observed under a confocal laser microscope.

## Statistical analysis

All experimental data were expressed as mean ± standard deviation. Statistical differences among the groups were analyzed using one-way analysis of variance (ANOVA). Statistical analysis was performed using SPSS (version 25.0; IBM, Armonk, NY, USA), with $P < 0.05$ considered statistically significant.

# RESULTS

### BMSCs-CM inhibited the expression of inflammatory factors in pro-inflammatory BV2 cells and promoted the expression of anti-inflammatory factors

The expressions of pro-inflammatory BV2 microglial markers (TNF-α and iNOS) significantly increased after LPS treatment ($P < 0.001$). After co-culture with BMSCs-CM, the expression of pro-inflammatory microglia-related markers decreased, whereas that of anti-inflammatory microglia markers (CD206 and Arg-1) increased significantly ($P < 0.001$; Fig. 1A).

The expression levels of inflammatory factors TNF-α and IL-6 were significantly increased in LPS-induced inflammatory BV2 cells ($P < 0.001$), whereas those of anti-inflammatory factors TGF-β and IL-10 were significantly decreased ($P < 0.001$). After co-culture with BMSCs-CM, the expressions of TNF-α and IL-6 were inhibited, whereas those of TGF-β and IL-10 were increased ($P < 0.001$; Fig. 1B).

### BMSCs-CM promoted mitophagy in pro-inflammatory microglia

Western blotting revealed that the expressions of mitophagy-related proteins LC3B and Parkin of pro-inflammatory microglia increased significantly after co-culture with BMSCs-CM ($P < 0.001$), whereas p62 protein expression significantly decreased ($P < 0.001$; Fig. 2A). The fluorescence intensity of ROS was significantly enhanced in LPS-induced pro-inflammatory BV2 cells, which decreased after co-culture with BMSCs-CM, indicating that BMSCs-CM inhibited ROS overproduction in pro-inflammatory microglia, thereby reducing the cellular oxidative stress response (Fig. 2B). Compared with the control group, the mitochondrial membrane potential of LPS-induced pro-inflammatory BV2 cells significantly decreased ($P < 0.001$). The mitochondrial membrane potential of BV2 cells was significantly increased in the LPS+BMSCs-CM group compared with

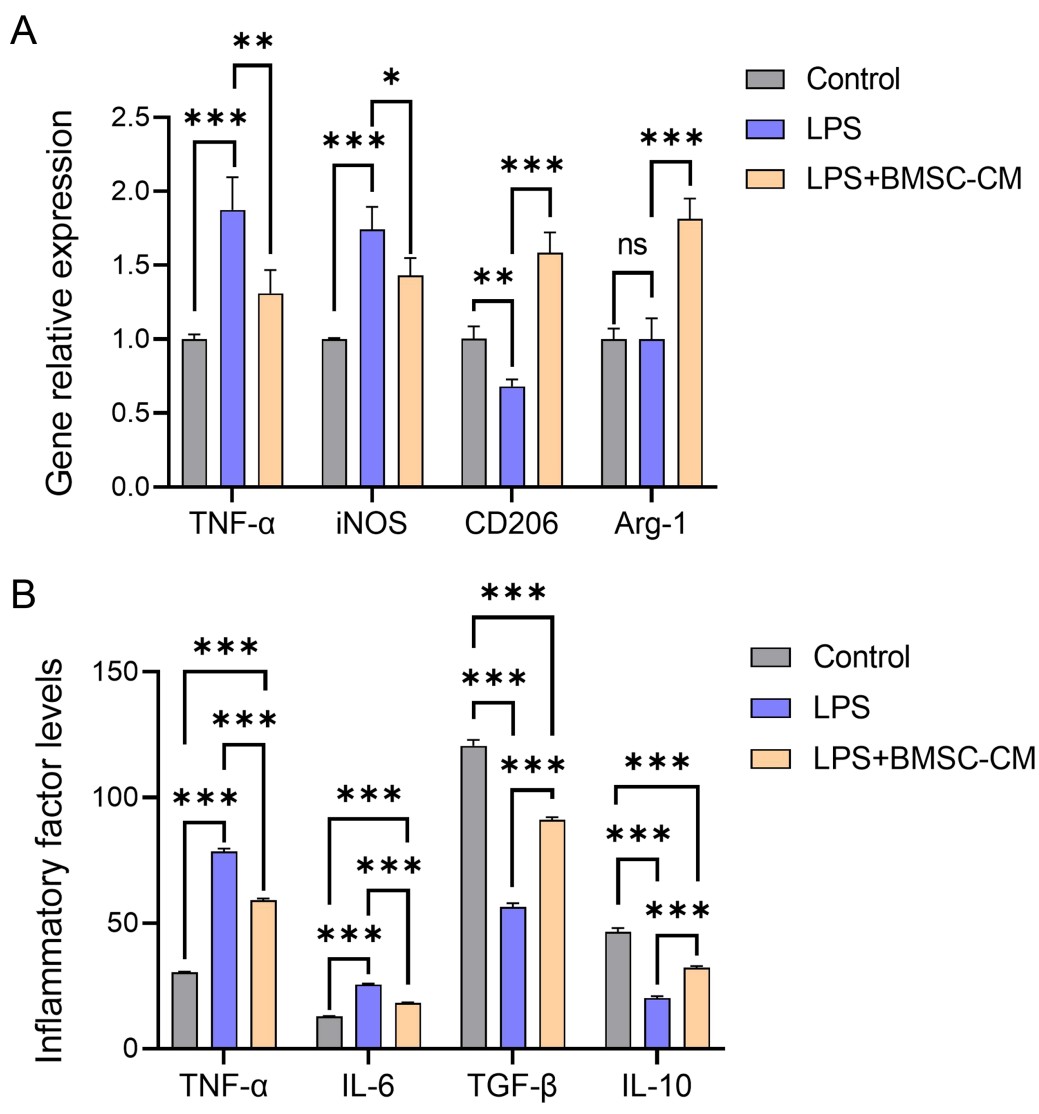

**Figure 1** (A–B) BMSCs-CM changed the phenotype and the expression levels of inflammatory and anti-inflammatory factors in BV2 cells.

the LPS group ($P < 0.001$), but it was still significantly lower than that in the control group ($P < 0.001$; Fig. 2C), indicating that BMSCs-CM partly inhibited the decreased mitochondrial membrane potential of pro-inflammatory microglia. Besides, the ATP content was significantly lower in LPS-induced pro-inflammatory BV2 cells than that in the control group ($P < 0.001$), whereas ATP content in BV2 cells was significantly increased after treatment with BMSCs-CM, but it was still lower than that in the control group (Fig. 2D), indicating that BMSCs-CM could promote the metabolism of pro-inflammatory microglia to return to homeostasis. Mitochondria and lysosome co-localization increased significantly in LPS-induced pro-inflammatory BV2 cells (Fig. 2E), which promoted

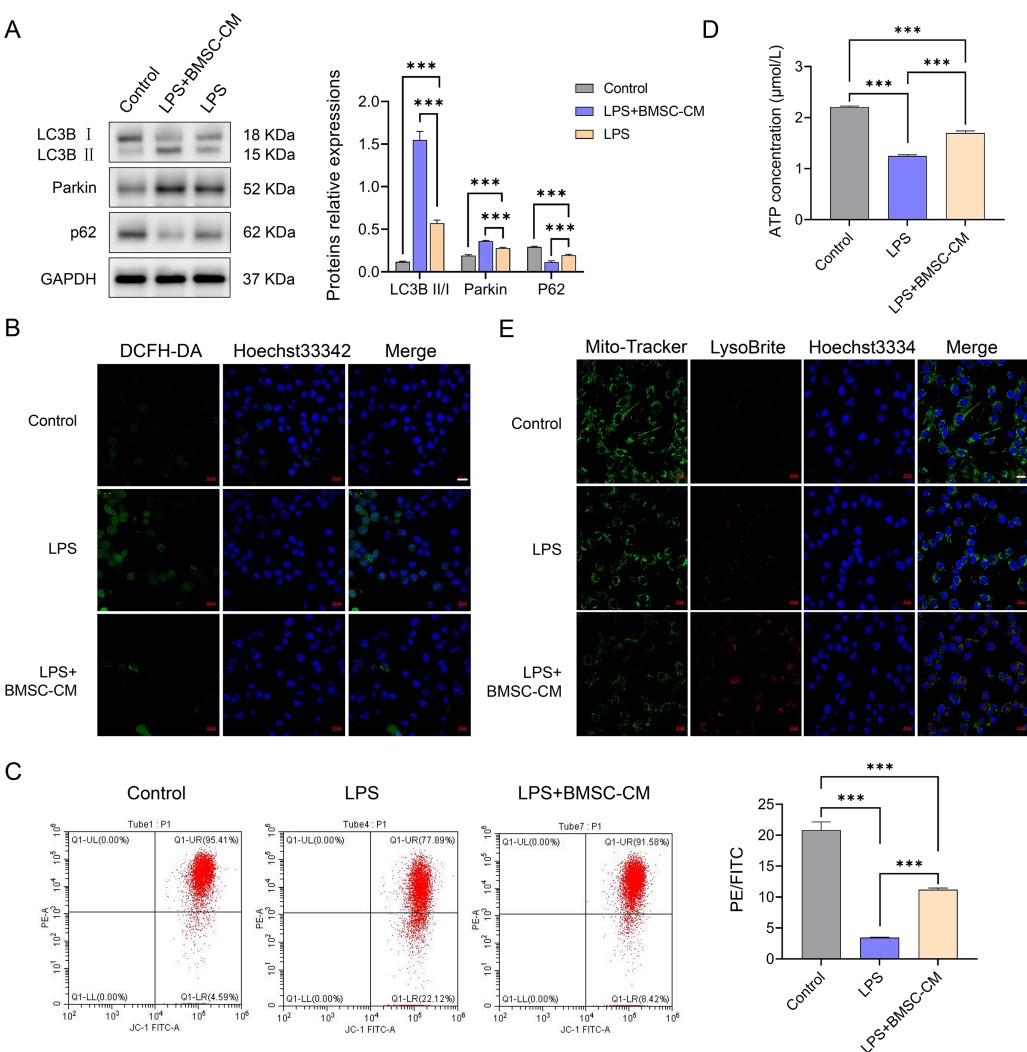

**Figure 2 Roles of BMSC-CM on mitophagy in LPS-induced BV2 cells.** (A) The expression of mitochondrial autophagy associated proteins was significantly increased after co-culture with BMSCs-CM. (B) After co-culture with BMSCs-CM, the fluorescence intensity of ROS in BV2 cells changed from increasing to decreasing. (C) After co-culture with BMSCs-CM, the mitochondrial membrane potential of BV2 cells changed from decreasing to increasing, but still lower than that of the control group. (D) After co-culture with BMSCs-CM, the ATP content of BV2 cells changed from decreasing to increasing, but still lower than that of the control group. (E) Colocalization of mitochondria and lysosomes increased significantly in LPS-induced BV2 cells, while the number of dysfunctional mitochondria decreased after BMSCs-CM co-culture and inhibited the overproduction of ROS.

mitophagy in pro-inflammatory microglia after co-culture with BMSCs-CM, cleared dysfunctional mitochondria, and inhibited ROS overproduction.

## Inflammatory BV2 cells co-cultured with BMSCs-CM regulate the proliferation, migration, and apoptosis of C8-D1A cells

Inflammatory BV2 cells were co-cultured with astrocyte C8-D1A, which significantly promoted the apoptosis of C8-D1A cells and inhibited the proliferation and migration of

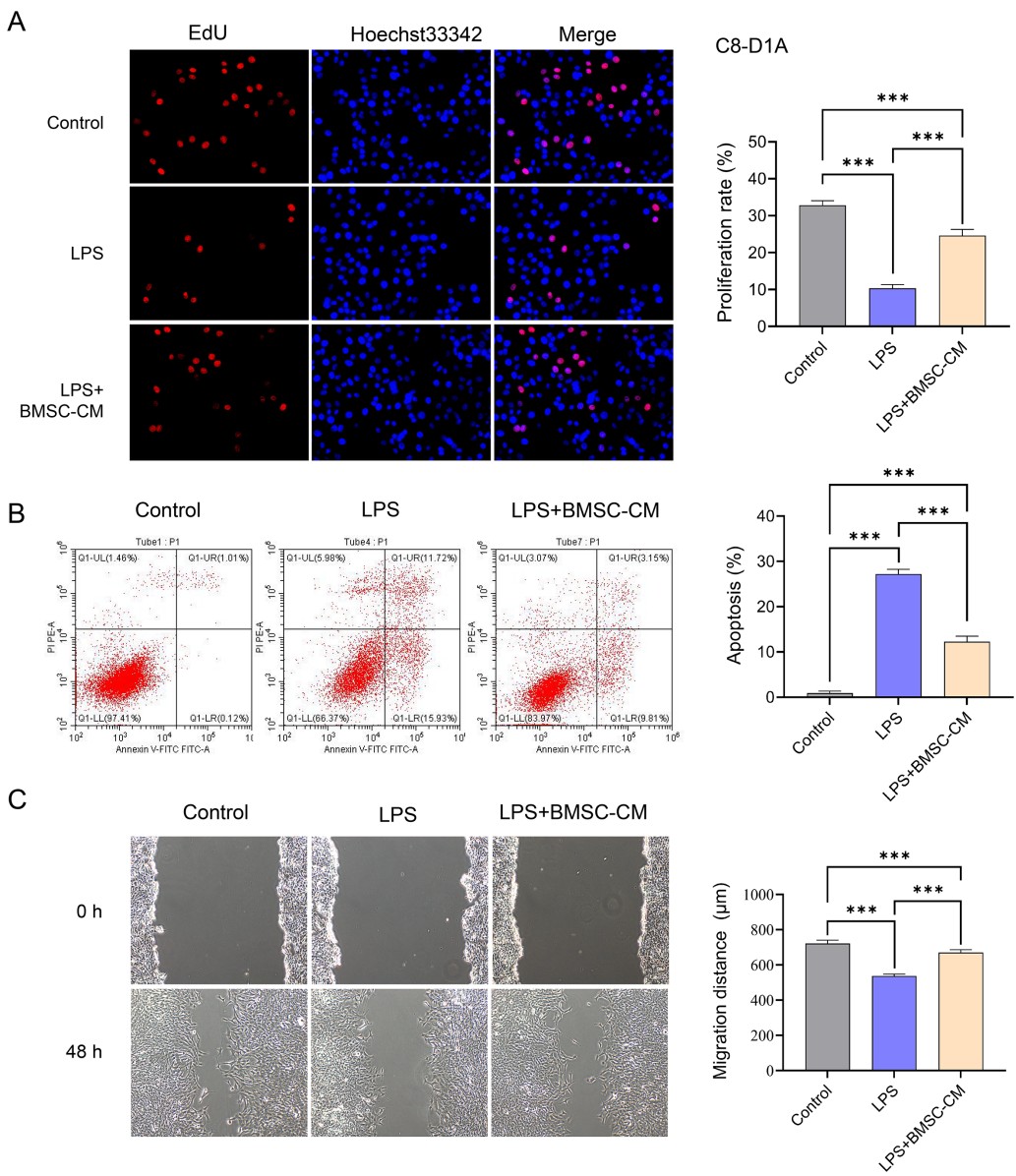

**Figure 3** **The effects of BMSCs-CM on C8-D1A cells.** Lps-induced BV2 cells significantly promoted apoptosis of C8-D1A cells (A), inhibited proliferation of C8-D1A cells (B), and increased the scratch distance (C). After co-culture with BMSCs-CM, apoptosis decreased (A), proliferation increased (B), and scratch distance decreased in C8-D1A cells (C).

C8-D1A cells. After co-culture with BMSCs-CM, the proliferation and migration activities of C8-D1A cells were significantly increased, and apoptosis was significantly reduced (Figs. 3A–3C).

## DISCUSSION

In the complex pathological changes following acute SCI, microglia, as sentinels of injury response, respond rapidly and undergo transformation into various subtypes, which are primarily classified into pro-inflammatory and anti-inflammatory types from a functional point of view. Throughout all stages of SCI, inflammation plays a crucial role, such as in clearing dead tissue and cells, with pro-inflammatory microglia being a significant driver of the inflammatory response after SCI. Numerous studies have examined how to control the extent of inflammation and prevent damage to normal nerve tissues and cells caused by inflammation; promoting the transformation of microglia from a pro-inflammatory phenotype to an anti-inflammatory phenotype is an important strategy (*Jiang et al., 2023a*; *Wei et al., 2023*; *Yang et al., 2023*).

Based on the findings of a previous study *Brennan et al. (2022)*, we successfully established a pro-inflammatory microglia model *in vitro* using LPS-induced BV2 cells. After co-culture with BMSCs-CM, the expression of pro-inflammatory microglia markers TNF-α and iNOS exhibited a downward trend, whereas that of anti-inflammatory microglia markers (CD206 and Arg-1) significantly increased, indicating that BMSCs-CM can facilitate the transformation of microglia from a pro-inflammatory to an anti-inflammatory phenotype. Correspondingly, after co-culture with BMSCs-CM, inflammatory factors TNF-α and IL-6, which were initially highly expressed in pro-inflammatory microglia, exhibited a significantly decreasing trend, whereas TGF-β and IL-10 transitioned from low to high expression levels. This indicates that after the transition of microglia from a pro-inflammatory to an anti-inflammatory phenotype, they exhibit certain anti-inflammatory functions, potentially reducing the deterioration and progression of the inflammatory response. The TGF-β signaling pathway has been reported to play a crucial role in neuronal differentiation, growth, survival, and axon regeneration following central nervous system injury (*Ma et al., 2024*). After SCI, regulation of the TGF-β signaling pathway can inhibit inflammation, reduce apoptosis, prevent glial scarring, and promote nerve regeneration. M2 macrophages in the SCI region activate the TGF-β signaling pathway by secreting TGF-β, resulting in increased expression of key blood-spinal cord barrier (BSCB)-related proteins, thereby accelerating the recovery of BSCB integrity and promoting functional recovery (*Nakazaki et al., 2021*). These studies suggest that BMSCs-CM may enhance the SCI process by stimulating the secretion of TGF-β by pro-inflammatory microglia; however, the specific mechanism remains to be further explored.

We further investigated the ROS content in pro-inflammatory microglia, as ROS overproduction is an important marker of cellular oxidative stress and a characteristic of the inflammatory response (*Xu et al., 2023*). The fluorescence intensity of ROS in pro-inflammatory BV2 cells was significantly enhanced, which decreased after co-culture with BMSCs-CM, indicating that BMSCs-CM can inhibit ROS overproduction in pro-inflammatory microglia, reduce cellular oxidative stress response, and alleviate inflammation.

Mitophagy is a self-protection mechanism of cells. When mitochondria function is abnormal, the autophagy mechanism is activated to eliminate dysfunctional mitochondria

and maintain cell homeostasis (*Jiang et al., 2023b*; *Wu et al., 2021*). To determine whether mitophagy is activated in pro-inflammatory BV2 cells after co-culture with BMSCs-CM, We evaluated the key proteins LC3B, Parkin, and p62 involved in mitophagy (*Kong et al., 2019*). The results revealed that the expressions of LC3B and Parkin proteins significantly increased, whereas that of p62 proteins significantly decreased during mitophagy, indicating that mitophagy was activated. Furthermore, we conducted in-depth studies on the process of mitophagy (*Mao et al., 2022*; *Meng et al., 2022*) and found that the mitochondrial membrane potential decreased in pro-inflammatory BV2 cells but increased after co-culture with BMSCs-CM; however, it was still significantly lower than that in the control group, indicating that BMSCs-CM promoted the mitochondrial membrane potential of pro-inflammatory microglia to return to homeostasis. ATP content is an important indicator of mitochondrial function status. In this study, ATP content in pro-inflammatory BV2 cells was significantly lower than that in the control group. After co-culture with BMSCs-CM, ATP content in these BV2 cells significantly increased but was still lower than that in the control group, indicating that BMSCs-CM can promote the metabolism of pro-inflammatory microglia to gradually return to homeostasis. However, the process may be prolonged. Colocalization of mitochondria and lysosomes is one of the main features of mitophagy (*Gu et al., 2020*). Increased colocalization of mitochondria and lysosomes was observed in pro-inflammatory BV2 cells, but co-culture with BMSCs-CM promoted the occurrence of mitophagy in pro-inflammatory microglia, cleared dysfunctional mitochondria, and inhibited ROS overproduction. By studying several key indicators involved in mitophagy, we found that although BMSCs-CM can activate mitophagy and promote the homeostasis of pro-inflammatory BV2 cells, these BV2 cells were significantly lower than in the control group, indicating that multiple other mechanisms may be involved in this process, necessitating further studies.

Although the results above are encouraging, we further studied whether BMSCs-CM impact the formation of glial scars. In SCI, the glial scar is considered a major physical barrier to neuron and nerve axon regeneration and nerve function recovery, with astrocytes being the main cells involved in glial scar formation (*Li et al., 2023*). Could the transition from pro-inflammatory microglia to anti-inflammatory microglia regulate the function of astrocytes and potentially reduce the formation of glial scars in the complex microenvironment in the body? Our study showed that when BV2 cells were co-cultured with astrocyte C8-D1A, pro-inflammatory BV2 cells significantly inhibited the proliferation and migration of C8-D1A cells and promoted the apoptosis of C8-D1A cells. After co-culture with BMSCs-CM, the proliferation and migration of C8-D1A cells were significantly increased, while apoptosis was inhibited. This is inconsistent with our assumption that the active anti-inflammatory response of microglia does not inhibit the proliferation and migration of C8-D1A cells and promotes their apoptosis, but rather had the opposite effect. Faced with this interesting phenomenon, we hypothesized that because microglia exhibit inflammatory characteristics after treatment with LPS, any cell in a stable state, including astrocytes, would appear impaired in the face of inflammatory agents. This also explains why pro-inflammatory BV2 cells significantly inhibited the proliferation and migration of C8-D1A cells and promoted the apoptosis of C8-D1A cells when the two were

co-cultured. After the co-culture of pro-inflammatory BV2 cells and CM, the inflammation of the former was weakened, and the damage of C8-D1A cells was also alleviated. Therefore, the proliferation and migration of C8-D1A cells were significantly increased, and apoptosis was significantly inhibited. This also indicates that although microglial cells can regulate astrocytes in a controlled environment *in vitro*, verification is still required in the complex *in vivo* microenvironment and to identify important related factors, which is our next step. It is worth noting that BMSCs-CM can regulate the phenotype transformation of microglia through mitochondrial autophagy, affect the proliferation and migration of astrocytes, and thus play a potential role in the formation of glial scars. Thus, this study provides a new perspective on microglia-astrocyte crosstalk and offers a novel direction for BMSCs-CM treatment of SCI.

## CONCLUSION

In summary, BMSCs-CM can promote the transformation of microglial phenotypes from pro-inflammatory to anti-inflammatory by regulating mitophagy and can also regulate the proliferation and migration of astrocytes. This study lays a theoretical foundation for BMSCs-CM treatment of SCI, but the molecular mechanism of BMSCs-CM regulation of microglial mitochondrial autophagy and the effectiveness of treatment *in vivo* remain unclear. In the future, we will collectively analyze the key genes and pathways of BMSCs-CM in regulating mitochondrial autophagy and explore the molecular mechanism of microglial homeostasis mediated by BMSCs-CM by regulating mitochondrial autophagy both *in vivo* and *in vitro*, specifically in the formation of glial scars and nerve recovery after SCI.

### Funding
The authors received no funding for this work.

### Competing Interests
The authors declare there are no competing interests.

### Author Contributions
- Hangyu Ji conceived and designed the experiments, analyzed the data, prepared figures and/or tables, authored or reviewed drafts of the article, and approved the final draft.
- Weiming Chu performed the experiments, authored or reviewed drafts of the article, and approved the final draft.
- Yong Yang performed the experiments, authored or reviewed drafts of the article, and approved the final draft.
- Xin Peng performed the experiments, prepared figures and/or tables, and approved the final draft.
- Xiaoli Song conceived and designed the experiments, authored or reviewed drafts of the article, and approved the final draft.

## Data Availability

The raw data is available in the Supplemental Files.

## Supplemental Information

Supplemental information for this article can be found online at http://dx.doi.org/10.7717/peerj.17664#supplemental-information.

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
