# Peer review of "Conditioned culture medium of bone marrow mesenchymal stem cells promotes phenotypic transformation of microglia by regulating mitochondrial autophagy"

_PeerJ, doi:10.7717/peerj.17664_

## Round 0.1 · original submission · Major Revisions

· Academic Editor

Major Revisions

We have received the comments from two reviewers. While both reviewers acknowledge the importance of this research, they also point out areas for correction. I look forward to following their suggestions and making changes. In particular, reviewer 2's comments are important. Please correct the points this reviewer points out.

**Language Note:** The review process has identified that the English language must be improved. PeerJ can provide language editing services - please contact us at [email protected] for pricing (be sure to provide your manuscript number and title). Alternatively, you should make your own arrangements to improve the language quality and provide details in your response letter. – PeerJ Staff

·

Basic reporting

The author aimed to figure out the mechanism under which conditioned medium of bone marrow mesenchymal stem cells could affect the transition of polarized microglia from pro-inflammatory to anti-inflammatory phenotype. The research meets the standard of being original primary research within Scope of the journal, while research question is well defined and meaningful, stating how the research fills an identified knowledge gap in the field of cytopathology.
The manuscript is clearly written in professional,unambiguous language,with structure conforms to PeerJ standards as well as discipline norm for clarity. Background shows the context in a clear way with literature well referenced.
The study used correct cell line, focusing on the evidence of gene expression, mitochondrial metabolism, changes on pro-inflammatory factors and anti-inflammatory factors in BV2 cells as well as the migration change of C8-D1A cell under the LPS or BMSCs-CM condition. I commend the author for their preciseness in study design, performing rigorous investigation to a high technical and ethical standard with methods described with sufficient detail and information to replicate .
The result showed that BMSCs-CM promots the transformation of the phenotype of polarized microglia from pro-inflammatory to anti-inflammatory by regulating mitochondrial autophagy, as well as regulate the functional status of astrocytes. All underlying data, which are robust, statistically sound and controlled, are provided to support the result. At the same time, Raw data were supplied with figure of high quality and well defined.Conclusions are well stated, linked to original research question and limited to supporting results.
The manuscript is well-written, but still there are minor amelioration that I’d like to suggest:

1. For a better understanding of the result showed in Figure 2A Relationship between P62 and PINK1 isn’t clear, which require further explanation in the result section.

2. More impact and novelty of the study needs to be added in the discussion section.

3. The Western Blot result in Figure 2A should be clearly labeled, eg.LPS, LPS+BMSC-CM.

4. Clarity of the images needs to be guaranteed for a better reference of the study result, eg. Figure 2B and Figure 3.

5. Grammatical error requires correction include line 54-57.

Experimental design

please see 1

Validity of the findings

please see 1

Additional comments

please see 1

Reviewer 2 ·

Basic reporting

The authors of the article “Conditioned culture medium of bone marrow mesenchymal stem cells promotes phenotypic transformation of microglia by regulating mitochondrial autophagy” have examined BMSCs-CM on polarized microglia phenotype. They have found that co-culturing of BV2 cells with BMSCs-CM led to decrease of the expressions of
TNF-alpha and iNOS, while the expressions of CD206 and Arg-1 were increased.
At the same time, mitochondrial (Mt) autophagy proteins LC3, PINK1, p-Parkin, Mt membrane potential and ATP production increased in pro-inflammatory BV2 cells, while ROS production and apoptosis decreased.
The study is interesting and the topic is relevant. However, authors conclude that “BMSCs-CM promote the phenotype of polarized microglia from pro-inflammatory to anti-inflammatory by regulating mitochondrial autophagy, and regulate the functional state of astrocytes.” I cannot see the evidence from the authors work about regulation of the functional state of astrocytes, as they only checked proliferation and migration of C8-D1A cells.
English should be checked throughout of the manuscript.

Experimental design

1) What is the evidence that BM-MSCc used in the article are stem cells? Have they characterized stemness of these cells? Are they actually mesenchymal stromal cells with small population of stem cells? Please clarify!
2) In method section, rows 90-94, I am not exactly sure how the authors measured “the absorbance (OD) of each cell at 450nm” Please clarify!
3) All experiments in the paper were done in vitro. There is no in vivo animal model of spinal cord injury and treatment with MSC-CM, corroborating the findings. That study limitation should be addressed and discussed! The authors have mentioned it only as their future study.

Validity of the findings

Mitochondrial autophagy is frequently called in a literature as mitophagy. To document mitophagy transmission electron microscopy (TEM) is necessary to show the ultrastructure of the organelle, the integrity of the inner and outer mitochondrial membrane (IMM and OMM), changes in mitochondrial size and mitochondria-ER interaction. Without that evidence results presented in the study are of limited importance. That should also be discussed.

Additional comments

1. In the Abstract authors should shortly mention what BV2 and C8-D1A cells are. In addition, method section is very long for the abstract. Method details belong to the Method section.
2) The Introduction is very short and insufficient. Microglia and astrocyte function should be shortly introduced as well as their involvement in spinal cord injury.
3) Role of TGF-beta should be discussed more, as TGF-beta is important in wound healing, angiogenesis and immunoregulation.
4) Resolution of Figures and images is insufficient, should be improved!
5) Figure 3A (the expression of LC3, PINK1 and p-Parkin in LPS-treated BV2 cells was significantly increased after co-culture with BMSCs-CM), should include lane description of Western blot image, molecular mass of detected proteins, number of experiments done and quantification graph.
6) The authors should also include scheme/diagram as Figure 3 that would depict mechanisms involved in BMSCs-CM induced microglial phenotypic change, mitochondrial autophagy involvement, and regulation of the functional state of astrocytes.

---

## Round 0.2 · Minor Revisions

· Academic Editor

Minor Revisions

The reviewers appreciate that most of their concerns have been addressed. However, reviewer 1 makes one final suggestion, which I agree with. Please respond to it.

·

Basic reporting

The authors have addressed most of my previous concerns. One suggestion is that authors should label PINK1 in the result and material part to avoid confusion.

Experimental design

I have no further recommendations here.

Validity of the findings

I have no further recommendations here.

Additional comments

I have no further recommendations here.

Reviewer 2 ·

Basic reporting

The authors have answered all my questions and concerns and improved the paper considerably. I have no further comments or questions for the authors.
Paper could be published in current version.

Experimental design

No additional comments!

Validity of the findings

No additional comments!

Additional comments

No additional comments!

---

## Round 0.3 · accepted · Accept

· Academic Editor

Accept

I confirmed that the authors have addressed all of the reviewer's comments.